# An Architecture for Service Integration to Fully Support Novel Personalized Smart Tourism Offerings

**DOI:** 10.3390/s22041619

**Published:** 2022-02-18

**Authors:** Andrea Sabbioni, Thomas Villano, Antonio Corradi

**Affiliations:** Department of Computer Science and Engineering, University of Bologna, 40126 Bologna, Italy; thomas.villano@studio.unibo.it (T.V.); antonio.corradi@unibo.it (A.C.)

**Keywords:** smart tourism, social sensing, sensors, e-tourism, distributed-architecture, Spark, Zenoh, FaaS, Elasticsearch

## Abstract

The continuous evolution of IT (information technology) technologies is radically transforming many technical areas and social aspects, also reshaping the way we behave and looking for entertainment and leisure services. In that context, tourism experiences request to enhance the level of user involvement and integration and to create an ever more personalized and connected experience, by leveraging on the differentiated tourist services and information locally present in the territory, by pushing active participation of customers, and by taking advantage of the ever-increasing presence of sensors and IoT (Internet of Things) devices deployed in many realities. However, the deep fragmentation of services and technologies adopted in tourism context characterizes the whole information provided also by customer sensing and IoTs (Internet of Things) heterogeneity and deep clashes with an effective organization of smart tourism. This article presents APERTO5.0 (an Architecture for Personalization and Elaboration of services and data to Reshape Tourism Offers 5.0), an innovative architecture aiming at a whole integration and deep facilitation of tourism service and information organization and blending, to enable the re-provisioning of novel services as advanced aggregates or re-elaborated ones. The proposed solution will demonstrate its effectiveness in the context of Smart Tourism by choosing the real use case of the “Francigena way” (a pilgrim historical path), the Italian part.

## 1. Introduction

The rapid and pervasive evolution of digitalization covering all aspects of life has drastically changed the field of tourism [1] so as to propose a new pervasive experience, more and more based on online services and information; moreover, that trend is expected to further grow in acceptance and offerings. In fact, we already see a wide variety of services, datasets, and platforms concerning and supporting tourism in many of these new different forms. In the past decades, tourism acquired a key role in the development and economic growth of many countries, and so it is in Italy. As stated by Eurostat [2], in the EU (European Union) area tourism is EU’s third largest socio-economic activity, representing around 10% of the EU’s GDP (gross domestic product). Moreover, five EU Member States are among the world’s top ten tourist destinations worldwide.

In Italy, more significantly than in other countries, tourism is in continuous growth and represents a vital contribution to the wealth of the nation. In fact, the total contribution of tourism to the Italian economy in 2017 was 223.2 billion euros, equal to 13% of Italian GDP and Italy was ranked the second destination for outbound trips made by EU residents within the EU, in terms of nights spent [2].

The fast increase of the tourism market is raising the need for a “Smarter” Tourism (Smart Tourism, or ST for short) more able to personalize and adapt customer experiences while creating a more culturally rich and even more sustainable offer.

To highlight the importance of ST in the sustainable development of a country, the European Commission launched the European Capital of Smart Tourism to stimulate the development and sharing of ST good practices. The European Capital of ST is an initiative to promote the integrated offers of innovative, inclusive, culturally diverse, and sustainable practices to tourism development by European cities [3]. Within that project, the European Commission has defined the concept of Smart Tourism as the combination of properties:**Accessibility**: To enable barrier-free destinations and enable access, regardless of age, cultural background [4], and physical disability.**Sustainability**: To protect natural resources of a city, reduce seasonality, and include local communities.**Digitalization**: To use digital technologies to enhance all aspects of the whole tourism experience.**Cultural heritage and creativity**: To protect and capitalize on the local heritage for the benefit of all stakeholders: the destination actors, the industries, and tourists.

The always-enlarging availability of information accessible through the network has modified the approach of visitors to the experience from an even structured and well-planned tourism offering to more dynamic and by-need ones. The development of ST will require not only a more *personalized experience* for tourists but also a more *dynamic service* proposal as key factors of the whole experience. Examples of dynamicity are modern apps that exploit geo-localization to retrieve more suitable local services in the locality of tourists and in a by need fashion [5]. That dynamicity further promotes a more personalized experience by using intelligent system recommenders: today the whole information about previous historical data and profiling plays a key role in ST decision-making processes [6]. Moreover, in the last years, the recent global pandemic has stressed further the importance of smart and dynamic tourism services based on geographic positions [7].

The pervasive diffusion of IoT and smart devices, connected with social sensing, represents an important accelerator to drive information retrieval toward service quality. In Social Sensing, the final customer can be actively and deeply involved in many ways, from contributing with her knowledge and sharing data gathered with smartphone and personal wearables, up to asking him to complete simple tasks while moving with her phones. Social sensing extends an already widespread and well-established series of techniques called crowdsensing [8] and has already been proposed to involve users in the process of data gathering [9]. Since initiatives of social sensing, crowdsensing, and crowdsourcing can play an essential role in the development of smarter tourism services, those initiatives are also coupled with incentivizing user participation via some forms of competition among users and via strategies of gamification. Tourism gamification extends strategies from game design and involvement strategies in non-game contexts [10], so to influence consumer engagement, customer loyalty, brand awareness, and user experience in tourism areas [11]. Examples of these initiatives include the usage of a scoring system related to customer action undertaken also rewarded with forms digital or material incentives and rewards.

As an example, in recent years many studies have proposed the use of social sensing to collect geo-tagged information and exploit them to identify tourism areas of interest [12], map tourist behaviors [13,14], compare and differentiate clusters of tourists [15], and discover and propose again noteworthy new places [16].

From the user perspective, the employment of ST combined with social sensing can provide better tailored information to tourists, about the quality and accessibility of a place, suited to their peculiar interests, either long-term or defined on the spot [17]. Examples are many, from the presence of barriers to the support of different languages in the service, from the current weather situation to the current mood of the entire group. In the city of Bologna, for example, there is an application to cancel barriers both in access to services and in mobility as demonstrated by projects like mPASS [18] or Kimap [19].

Additionally, we add that the tourism area itself asks for deep integration with many other fields, such as smart cities [20], smart transport, smart wealth, and relative services and data sources [21] as a few examples of connected areas.

To summarize, the integration and combination of ST information and services are expected to create great business value [22] and to enable the development of smarter tourism services and experiences, but the heterogeneity of formats and interactions protocols slows down the integration of multiple platforms, making impossible the acceptance of a unique and comprehensive standard, because of the different stakeholders and organizations proposing ST services and the lack of cogent regulation [23].

To solve these problems, the authors propose APERTO5.0, a reference Architecture for Personalization and Elaboration of services and data to Reshape Tourism Offerings 5.0, based on human-centric interactions and data gathering favoring a strong personalization with interested tourists (either single, in groups of interests, or in other aggregation forms). The aim of APERTO5.0 is not only to exploit better existing resources for a greater business value but also to stimulate the creation of novel tourist services toward the whole potential of aggregation and augmentation of the platform. APERTO5.0 can present local services per se, but also present new offerings, composing together local services chosen depending on specific tourist profiles and current needs. In fact, the APERTO5.0 aims both to enable the creation of a platform that can present a unified view of services and information to tourists and to offer a single access point for advanced and augmented information and facilities composition to third parties private and public organizations, so to encourage and enhance the development of positive and sustainable tourism offerings.

In collaboration with some local companies, the authors have implemented the first prototype of APERTO5.0 to handle a specific scenario of tourism paths (“cammini” in Italian or ways in English). “Tourism paths”, also called pilgrim ways, are recognized routes connecting some historically relevant sites, and traditionally followed by pilgrims in the past: examples can be the “way of Saint James” to reach Santiago de Compostela, or the “Francigena way”, an itinerary going to Rome by foot, both started since the Middle Age. Those tourism experiences are gaining more and more attraction as able to offer a healthy traditional and sustainable experience, in line with tradition and connected with the territories traversed by the path.

To validate the capabilities of APERTO5.0 architecture, the authors have extensively tested the proposed solution under different loading conditions, simulating the “Francigena way” behavior in the Italian final part, by testing the possible occurrences in the different conditions of service request typically experienced in tourism real scenarios. The paper compares the results for the presented solution in gathering and requesting information with the solutions that represent the de facto standard of the market. APERTO5.0 showed a very good capacity of distributing and requesting information under different loading conditions and diverse network and infrastructural topologies complexity.

In summary, this paper significantly advances the state-of-the-art literature in the field with the original contributions listed below:(i)A novel architecture based on planes (see Section 3: Business, Cross Cutting Concerns, Data and Service) and Layers (Monitoring, Auditing Authorization Authentication, Presentation, Data, Analytics and Processing, Blending and Integration), to abstract the aspects related to the technologies used and group the different tasks.(ii)A practical approach to address the problem of heterogeneity and dissemination of information and services in the context of tourism services.(iii)An original implementation of the proposal based on open-source projects combining well established platforms with cutting-edge technologies.(iv)An application of the proposed architecture to the real-use case of “tourism paths”.(v)Some in-the-field experimental results for simple deployment cases to show the feasibility of the proposed approach and the efficiency of the implemented architecture.

## 2. The Proposal

APERTO5.0 is based on an organization that put together on the one hand all possible information sources, and services toward a better integration, on the other hand, the best proposition possible for the differentiated needs of all tourists, either single or in different composition groups in number and interests. The proposed architecture has been designed driven in the middle of existing tourism services and information providers and the possible requests and needs of customers (Figure 1). APERTO5.0 aims at becoming the reference for the development of new smarter tourism services and platforms while adding value for both producers and consumers based on its integration and augmentation capacity.

The architecture considers a producer every entity that provides information or services potentially appealing for the tourism field and its development, by including partners belonging to the public and private sector, open data, and any connected things spread in the interested region, like connected transports, sensors, and user wearable devices. The absence of a common agreement and regulatory organs leads to a wide specter of interaction modes and formats, as well to many heterogeneous types of agreement and relationships proposed by different producers. On the other side, customers are users of the platform that can consume services and data resulting from the processes of augmentation, elaboration, and orchestration of information and services coming from Providers. The authors target does not consider only tourists as customers of the platform, but also municipalities, destination managers, third parties tourism services, and all the other actors possibly present in the territory and interested in participating in the formation of a smart tourism offer. Such heterogeneity in customers leads to the requirement of a high level of modularity and adaptability to suit the different needs of customers.

We must stress that a provider can also play the role of customer and vice-versa, by creating a circular Prod-Cons pattern, where providers can interact with the proposed platform as customers to grow their services, and customers can improve their experience through personal contributions to APERTO5.0. This positive evolution can be opportunistically encouraged through initiatives, such as crowdsourcing campaigns and the creation of local relationship networks. Authors claim that the introduction of the proposed digital platform can encourage the creation of a network of partners that can also increase, monitor, and guarantee the value of data and services. These types of relationships can also enable the development of opportunities by creating a mutual value, in the sense of social results in the tourism field, such as the creation of an agreement of multiple municipalities crossed by a path of cycle tourism. The network constituted with the different partners in the territory supported by the proposed digital platform constitutes the target supply chain for customers.

The architecture of APERTO5.0 is based on three well-defined layers called planes: the higher-layer business plane is responsible for the interaction with customers; the other two lower-layer planes are responsible for all possible services (service plane) and information (data plane) arriving from interested providers. It is important to stress that APERTO5.0 can expose new services based on the available existing ones; another lower layer component is the crosscutting one, in charge of all managing and monitoring functions of the entire architecture.

We are now expanding the details for the above planes. The business plane is the functional plane that addresses the complexity derived from interactions with customers with the main goal of providing a unique point of interaction, and, at the same time, of hiding from the customers the complexity of distributed datasets and services. The business plane has the main goal of uniform access to the heterogeneity of services, protocols, and interactions arriving from the underline planes to compose a solution offer. This plane drives the composition, coordination, orchestration, and exposition of services and information coming from the data plane and the service plane. This plane is capable of creating new synthetic tourism proposals, starting from existing services and information, such as in the creation of new packet experience, by booking public transportation, and by creating a path across most visited places. These features demand a high level of modularity and composability to adapt to the continuously evolving needs and interaction methods of customers, via tools like Dashboards, Apps, and APIs.

The data plane is the component of APERTO5.0 responsible for collecting, managing, and analyzing all the datasets and information collected from third parties providers, realizing the augmentation and conformation of data. This process is an essential step in the creation of new services and platforms, so that uniformity and standardization can reduce considerably the effort related to the management of different formats. The data plan can handle and process both data in motion and at rest (very static and very dynamic data, as extremes), enabling the exploitation of both historical data and fresh real-time information. To achieve a good value from data both stored and processed, it is necessary both the use blending techniques, over the data coming from multiple and diverse sources to merge them and the consolidation of a network of partnerships in the territory that can provide feedback and support, so as to specifically verify the information. The data plane supports multiple types of representation and analytics, in order to handle the different needs of customers, including geographical and time-based queries, up to computationally intensive processing like graph algorithms and machine learning techniques.

These planes, part of the proposed architecture, enable the representation, integration, and orchestration of provider services. Since the proposed solution does not replace or force migration of existing services but, on the contrary, focuses on empowering existing ones, the service plane has the goal of matching to each existing service one or more *synthetic* services representing it internally to the platform. Each *synthetic service* handles all the specificity of the target producer service like protocol, billings, and authentications taking charge of all the necessary coordination with other planes present in the proposed solution, specifically the *cross cutting concern plane*. Moreover, the services can be further composed or decomposed to create new *synthesis* services, so to create offerings at different granularities, e.g., a transportation service can be composed starting from a sharing mobility service and a public transport one. To enable these advanced techniques, the service plane also introduces a series of composing categories applicable to services that can not only enable a simpler composition of services but also simplify the suggestion of alternatives to the final customer, e.g., to suggest alternative places to visit, transportation to reach a point, or nearby hosting structures. Furthermore, these types of abstractions can enable smarter behaviors exploited by customers, leading them to a better tourism experience capable of reacting dynamically to events and information received, e.g., a signal of breakdown of a bus can trigger the automatic call of an alternative transportation partner or the proposal of remediation proposing like free hosting structures.

Finally, APERTO5.0 defines a *cross cutting concern plane* (CCCP) consisting of a series of components to implement all cross-cutting concerns and support other planes, in the whole management and interaction with internal and external services. Authors expect a continuous evolution of the CCCP while dealing with new challenges and new scenarios, especially distributed across a heterogeneous territory like the one covered by a pervasive platform of tourism. The CCCP supports the resolution of problems not only addressed internally to proposed infrastructure but also directly in relation with customers or providers, e.g., health checks of provider services and endpoints. Some services belongings to the CCCP can not only support other services but can become themselves part of the offer of other planes, e.g., the authentication services that can be provided as a service directly to customers behaving as part of the business plane. The components realized in the CCCP layer aim at operating transparently to the other component of the infrastructure. In this way, the evolution and introduction of new features in the CCCP plane can benefit, with little effort, multiple components belonging to other planes of the proposed solution.

## 3. APERTO5.0 Architecture Full Component Description

Going deeper into a more detailed description of APERTO5.0, we magnify the presented solution that is fully partitioned into detailed layers, each one corresponding to a single business process (Figure 2). We describe here: the presentation layer that implements and proposes a unique view of all services within the business plane; the blending and the data layers inside the data plane that allow the input of all information needed by ST by polishing and presenting, and also storing within the second component; the analytic and processing layer together with the *integration* layer constitute the service plane, where the former is capable of extracting any possible interesting service from the proposed available ones, while the latter is capable of getting to all available services available for ST. In addition, the *auditing, authentication, and authorization* (AAA) and the *monitoring* layers constitute the first two proposed modules that realize the cross cutting concerns plane.

The presentation layer constitutes the main component of the business plane to create and provide a unified view for tourism and third actors, by combining information and services coming from data plane and service plane to provide new smarter tourism services, typically the creation of a travel experience blending information on events and places in the destination with services of ticket purchase for transport. Supported customer interaction can employ heterogeneous technical protocols, e.g., pub-sub, client/server, fire and forget, and advanced query languages.

The data plane is subdivided into two horizontal layers: the blending layer and the analytics and processing layer (AP Layer) and one vertical layer: the data one.

The *blending layer* is responsible for gathering, cleaning, and adapting to a convenient format the information coming from third-party services, open data, and custom ad-hoc services. These data sets are then stored, according to predefined or dynamic policies on the different storage services composing the data layer. This lower layer implements the integration logic with the different forms of interactions and queries exposed by external data sources. The interaction methods are managed and adapted to require no changes on the provider side and can support reactive interaction, such as event-based traffic information systems and scheduled/polling-based ones like information about shows, fairs, and festivals periodically published on a site. The blending layer widely exploits principles of modularity and composability, and any introduction of new data sources or data manipulations follows the plug-in logic with a minimum effort of development and instantiation, enabling in such a way a sustainable growth of handled producers. This component implements a gathering approach direct from producer sources, in this way enhancing diversity and customized experience. This approach differs significantly from more traditional ones such as in booking.com [24], where the hosting infrastructure must register and constantly update its own data in a third-party portal, so requiring a standard format a-priori.

As part of the data plane, the *analytics and processing layer* aggregates and analyzes the different data sources stored in the data layer and exposed by the integration layer. This layer realizes the process of adding value to the information coming from local tourism offerings, via an internal creation of new aggregate datasets and the discovery of new insights through advanced elaboration techniques, such as big data processing, data mining, and AI (artificial intelligence) algorithms. This layer can also support real-time event-based and continuous-stream processing to enable advanced real-time queries and subscription mechanisms, as well batch operations for heavier time-consuming analytics.

The *data layer* is a vertical layer inside its plane, since it cooperates with all other layers, by storing elaborated data, schemes, and metadata and by exposing them through advanced indexing and query languages. The data layer handles the storing of fresh and past collected datasets and metadata, by enabling fast and advanced analytics and interactions through the exploitation of the data locality principle and advanced indexing, and by proposing customer-adapted viewing techniques. To support an effective memorization and query system the data layer exploits the most convenient storage strategy, so supporting any different data format and memorization technology.

The *integration layer* cooperates with the *analytics and processing layer* inside the service plane. The integration layer, in particular, is responsible for re-exposing in a convenient and optimized way the external services provided by third parties. The exploitation of many different categories enables the possibility to combine properly and substitute service calls to create smarter and reactive services, e.g., buy theater, cinemas, or public transportation tickets. This wrapping mechanism enables to hide and abstract from the peculiarities of each service, such as internal protocol, service call sequence, or rate-limiting, and facilitates the coordination and combination realized in a analytics and processing layer. Moreover, this layer interacts with all external services and datasets to obtain data not directly available, since they are filtered away, such as real-time number of available tickets or current position and updated time of arrival of a transport. The integration layer to support integration with different heterogeneous providers implements many types of interaction including periodically and reactively.

The *auditing, authentication, and authorization* (AAA) and the *monitoring* layers constitute the core of the cross cutting concerns plane, available to all other components, to operate in conjunction with all layers in the proposed architecture. These two layers also form an important part of the business plane as they provide important services to the final customer e.g., authentication service or metric.

In fact, the AAA layer is not only responsible for guaranteeing a proficient level of security to the infrastructure layer but also to provide a unique point of access, for final users, to the services covered and integrated into the platform. This allows to preventing registration and policy adaptation to any tourism service provider and enables a unique view for tourists and third-party organizations.

The *monitoring layer* provides useful insights on the service usage and the overall state of the platform to enable both elastic management of the infrastructure and significant added business value. In fact, from the monitoring layer it is possible to extract and underline trends in services usage with a geographical and temporal connotation and exploit them internally at the platform or supply “as they are” to external organizations, think to the trends in ticket buyout of a public transport localized in a determined time or region. Moreover, this layer can control malfunctions and unavailability of services and information provided by producers, by generating alerts by need to request automatic execution of recovery action.

The complexity and stratification of the proposed architecture derives from the exigence of addressing heterogeneity in tourism information and services as well as in customer needs. Moreover, this complexity is expected to further increase with many additional layers and components while increasingly addressing more and more ST use cases with their intrinsic exigences. Authors claim, however, the validity of the proposed architecture as a base for the structuration of a solution able to satisfy and integrate many ST vertical scenarios.

## 4. The Case Study of Tourism Paths

One of the most challenging scenarios in the context of smart tourism is the business of tourism paths (or ways or itineraries, sometimes pilgrim’s ways), typically established very long ago to suggest routes to religious pilgrims in the middle age and to give advice in their ways toward their final destination. This paper takes the “Francigena way” as a use case, in particular, the Italian part passing through territory. This novel and more requested type of tourism offers are characterized by the requirement of extreme personalization and dynamicity of target user experience; tourists can choose via the information given by their smart devices how to continue the experience, which is always driven by current information derived from their connection on demand. Of course, the same always-connected feature can apply to many other areas apart from ST, so presented architecture can be crucial in those too.

In tourism paths, users intend to use ICT as an essential part of the experience and tend both to be driven by information they need to get either personally or as part of a group, and to feed information over the community depending on their current experience. We consider that tourists can become prosumers (consumers and producers at the same time) of the experience of tourism paths. As an example, via ICT tools, the user can interact more dynamically and satisfactorily, by choosing to read personalized paths and calibrating languages and contents based on specific levels of learning [25]. It becomes essential to gather as much information as possible, so as to provide customers with the necessary details and to provide the customers with the best experience possible. That high dynamicity constitutes a challenge for providers of smart tourism services characterized by huge distances covered, with a multitude of information gathered, with a high fluctuation in number of users requesting those services, and with an important level of heterogeneity in partners to be involved.

In collaboration with Imola Informatica, authors then decided to create a first prototype of APERTO 5.0 to support the creation of a platform of smart tourism for tourism paths. This platform aims to support customers in all the phases of the experience: from the *planning of the trip*, also *during the experience*, and finally in the *post-experience phase*.

From the user point of view, a basic service example should make it possible to (1) plan the trip in any plan details, such as choosing also where to sleep and eat; (2) contacting local shops and realities for the discovery and purchase of typical products; (3) explore the path you want to take, discovering what to visit. That is normal in the planning phase, but those services should be available also during the experience itself, and at the same time, must occur in the post experience, both to document the events and compare with possible different expectations for future planning.

Particular attention is devoted to the process of information gathering from partners, sensors, and user contributions. Users are continually encouraged to contribute during the whole experience with images, gathering data from sensors, expressing opinions, and sharing their GPS tracks.

This process of social sensing opens the APERTO5.0 platform to a continuous enrichment of the experience introducing in the ST offerings new variations to the route, points of interest, and information about places acquired by other users during similar experiences. Moreover, the entire information allows the platform to dynamically adapt customer experiences, to make the offer more accessible and enjoyable. In the scenario of tourism path, a parameter to be continuously monitored is the *difficulty* associated with a stretch of the route that depends on many environmental current factors, such as humidity in the air and the soil, condition of the track, and wind force.

To stimulate user contribution, the platform exploits a gamification approach, where each user contribution assigns some score points according to the relevance of the intervention. As an example, the signaling of a critical problem during the tourist path (particularly important for other users) can be estimated as a relevant contribution, while the review of a restaurant can be associated with fewer points. At the overpassing of some predefined thresholds, the user is rewarded with some prizes of many diverse types and values, such as showing the user status and coupons offering discounts on partners related to the experience.

## 5. Materials and Methods

To evaluate the potentiality of the presented architecture proposal the authors have developed a first implementation prototype based on some widely open source affirmed platforms, with the goal of not only helping us to evaluate the effectiveness of the proposal in some real use case scenarios but also discovering whether current technological solutions can fulfill the use case needs. As shown in Figure 3 we base the first prototype on 5 main technologies: (1) Eclipse Zenoh [26], (2) OpenFaaS [27], (3) Apache Kafka [28], (4) Apache Spark [29], (5) Elastic Stack (Elasticsearch and Kibana) [30], necessary to satisfy the needs of the “Francigena way” use case and presented from a bottom-up perspective from the data gathering to the final service and data representation to customers.

Potential technological gaps discovered during the execution of the testbed will constitute further directions of research to address and solve the problems of a general architecture and infrastructure to support the development of smart tourism.

One of the most challenging tasks demanded of the proposed infrastructure and demanded to the blending layer is the gathering and dynamic querying of information from different sensors, customers, and providers present in the territory. The difficulty of this task is further exacerbated by the use case scenarios that must cover a large territory with very heterogeneous characteristics, like connection quality, coverage, and density of devices. To overcome these challenges, we have introduced Eclipse Zenoh as the preferred interaction medium between the infrastructure and data sources present along pilgrim’s paths. Zenoh is an open project developed by Eclipse Foundation born to fulfill the need for an efficient adaptation of fog-centric business. In the past decades, the cloud-centric model has taken hold in several fields and has been applied pervasively to countless business cases. Nevertheless, the cloud-centric model has shown some limitations under some application conditions and fields (e.g., limited connection bandwidth, low latency, negligible connection cost, etc.). Zenoh tries to fulfill those needs, by implementing a distributed pub/sub model with a limited footprint and overhead to reduce to the minimum the impact over the business resources. At the same time, Zenoh aims to keep as low as possible the latency and to increase to the maximum throughput.

Each node that uses the Zenoh APIs is declared to belong to one unit type: peer, client, and router, since these three units are the building blocks for any application. In fact, the application unit declared as *peer node* could connect to another node in the network allowing a complete graph topology or could select the nodes to connect to, to form a connected graph topology. The role of the *client node* is to connect to a single router or peer to communicate to the rest of the system. The *router node* is a software process able to perform level 7 routing (OSI model) and connect to different nodes to connect different topologies to each other to form any kind of topology.

Zenoh also provides and implements a configurable discovery mechanism that employs a gossip protocol over multicast, to auto-connect to the other nodes with the aim of using a so-called “path” to perform a request. The “path” is a string that works as a key thus acting as a symbolic location that expresses the logical function rather than IP addresses so that this organization allows the application to hide the exact position of the data enabling a geo-distributed storage implementation. To enable the gathering and dynamic querying of data across the distributed infrastructure, Zenoh provides a set of high-level APIs that hides the underline infrastructure complexity, while enabling advanced and well-established communication patterns, like publish/subscribe or request/reply. Those APIs together with the advanced infrastructure of Zenoh make possible for APERTO 5.0 the collecting of information from the action of user sensing and the contributions of partners, as well as the dynamic querying of data at rest present in the wide territory covered.

To address the great variability of sources in a Prosumer context, the authors introduced a layer of adaptation and abstraction at the forefront of the proposed architecture and realized by OpenFaaS an Open-Source Function as a Service (FaaS) platform. FaaS computing is a novel model of cloud computing that promotes the absence of customer control over the infrastructure by specifying only the creation and upload of the desired business logic. This model can enable an unprecedented speedup in the development of new components, such as the creation of ad-hoc components to manage and adapt data coming from various sources.

Moreover, the FaaS model is characterized by a fine-grained scaling of resources associated with each service. In fact, in FaaS platforms, the code representing the business logic is not always active and in execution but is dynamically executed by the platform only at the arrival of the user-associated events. That enables a good consolidation of different services deployed on the same hardware, potentially achieving an important cost-saving.

Finally, recent developments in FaaS, such as function chaining or map reduction, aim to introduce even more capabilities and paradigms support that can promise the standing of FaaS platforms as a crucial component to forefront complexity of tourism cases. The FaaS model paves its interaction model and scalability on asynchronous communication and stateless computation, while modern FaaS platforms, such as OpenFaaS, still lack mechanisms of synchronization and aggregation. Therefore, we had to recur to one well-spread message-oriented middleware, Apache Kafka, as a layer of interaction between functions and other functions or integrated platforms. Apache Kafka is a highly scalable, open-source streaming platform that provides the *Pub/Sub protocols*. We opted for Apache Kafka, among many open-source MoM present on the market, for its capacity of excellent scaling in handling a massive number of concurrent messages with multiple configurable qualities.

In fact, the FaaS layer can potentially create a large amount of information and as the importance of information can significantly vary, so a differentiation of QoS support in the service is essential. Data coming from the integration and blending layers are then convoyed as a stream in specific Kafka topics each one characterized not only by a business means but also by a specific QoS. The continuous stream of differentiated information can be either kept in persistent storage or streamed to the analytics and processing layer for further processing. Kafka enables not only to handle a huge amount of concurrent information coming from different providers but also to handle diverse sources with diverse levels of QoS, spanning all common policies, from best effort to atomic delivery [31].

To address the continuously evolving needs of customers we introduce in the processing layer Apache Spark, an open-source distributed analytics engine memory computation large-scale data processing that provides multiple modules like MLlib, GraphX, Structured Streaming to enable the parallel and in-memory computation of data at rest and streamed from Kafka with advanced techniques, including graph processing, machine learning, incremental computation, and stream processing [32]. Inside the proposed solutions, Spark is the essential component in charge of processing and interpolating data coming from sensors, social networks, and social sensing. For specific partner requirements, we can opportunistically open the possibility to directly exploit the advanced processing capabilities of Spark. The process of submitting a task to Spark, however, is not trivial and highly depends on the characteristics of the deployed environment. For this reason, we decided to provide a layer of abstraction through Apache Livy [33] to enable the direct submission of Spark Jobs via REST API. Furthermore, Livy also takes care of implementing politics and mechanisms to guarantee fault-tolerant submissions and concurrent request support in a multi-tenant environment.

All the information elaborated by Spark and the FaaS platform are eventually stored in Elasticsearch, a persistent, distributed, and fault-tolerant NoSQL database. Elasticsearch is a distributed and scalable open-source project part of the ELK Stack. Through its REST APIs Elasticsearch enables the store and advanced analytics and query of big volume of data in near-real time fashion.

In the data layer, in fact, two main needs to handle are the management of a huge amount of data and the creation of a query engine service with advanced capabilities to support visualizations and customer queries.

The introduction of Elasticsearch in proposed technological stack enables us to maintain relevant historical information while creating efficient and practical indexes and views to facilitate the retrieval of data. Data so memorized can be queried by customer through API or presented as advanced graphs, dashboards, dynamic maps in Kibana an open frontend service part of the elastic stack providing search and data visualization capabilities for data indexed in Elasticsearch.

The proposed architecture paves its strength on the modularity of the solutions composing the different layers. The different components proposed, in fact, while suiting the needs of the “Francigena way” cannot match the requirements of other ST use cases or result in overkilling and can be replaced accordingly. We expect, indeed, a great and continuous evolution in components of the single layers of the architecture presented, according to the increasing number of addressed use cases.

## 6. Results

The prototype of APERTO5.0 constitutes the backbone of services where the different partners constellating the tourism path and tourists receive and provide information also exploiting ad-hoc developed mobile-oriented applications.

Given the huge and ever-increasing number of data sources that APERTO5.0 is required to handle we are interested to present here two main quite common and potentially critical processes: (1) The action of gathering and querying data across complex and distributed topologies of sources; and (2) the processing and integration of cases to demonstrate the ability of proposed platform of gathering and process data and services in order to be re-exposed. We concentrate on these two specific functions, since the huge quantity of data produced by geographically distant places in the “Francigena way” requires a powerful and fast gathering processing phases, so as to maximize the value of the obtained information.

The tests are organized along the two major problems highlighted: with the first test-bed section aiming in proving the ability of Zenoh in supporting presented architecture in the phases of receiving and querying of data and with the second section with the goal of confirming the capacity of the chosen platform in adapting and processing incoming information. Afterward, we confirm the introduction of a FaaS platform as an essential part able to address the extreme variability in tourism service load while also facilitating the fast development of connectors to take on the Prod-Con heterogeneity.

The tests of the first testbed section are deployed on an infrastructure composed of 5 nodes equipped with an Intel(R) Core (TM) i5-3470 CPU running at 3.20 GHz and 8 GB of RAM each. All the tests of the second testbed section have been conducted on a physical infrastructure composed of five virtual machines each equipped with 8 vCores, 32 GB vRAM, and 150 GB data SSD.

### 6.1. Social Sensing Data Gathering

Infrastructure topologies interconnecting prosumers in the context of pilgrims paths can assume diverse levels of complexity by traversing many networks and applicative endpoints. This complexity is further worsened by the fact that a user will consume in their experiences not only local data but potentially data distributed far away, many stages before or after the current location. For this reason, this section aims at testing the capabilities of Zenoh in supporting APERTO5.0 in both processes of receiving data through a standard Pub/Sub protocol and querying data at rest present in separate locations, with varying interconnection topologies and load. We then compare the obtained results with two often adopted technologies for data gathering, namely RabbitMQ [34] and a cascade of Envoy HTTP Reverse proxy under the same conditions [35].

In the use case of “Francigena way”, one of the most challenging tasks that proposed infrastructure requested to realize is the ability to gather and retrieve data traversing arbitrary complex networks and infrastructural topologies. This happens often during social sensing action where the data coming from sensors are requested to traverse different devices like user smartphone or Wi-Fi Access Points compared to the ability of Zenoh, RabbitMQ, and HTTP of traversing multiple hops to interrogate sources of data in the territory.

We have triggered an increasing number of requests starting from a frequency of 1 request per second and reaching 1000 requests per second. Results shown in Figure 4 highlight that in a request/reply interaction HTTP performs better when the number of hops traversed increases.

We then tested the ability of the different solutions to satisfy higher rates of requests experienceable when multiple users and devices send contemporary a lot of data. We then repeat the precedent experiment in the case of 2 hops to be traversed and we sent an increasing rate of requests starting from 0 to 10,000 requests per second.

As Figure 5 shows, while HTTP forwarding realized through proxies performs better at low request rates when the rate exceeds 4000 requests per second the performance of this protocol starts to degrade with a fluctuating behavior and an RTT measured higher than once of Zenoh and RabbitMQ.

We then tested the ability of RabbitMQ and Zenoh of distributing and collecting information through a typical Publish/Subscribe mechanism. This test foresees the exclusion of the HTTP Proxy and HTTP protocol does not support such type of interaction.

In particular, the third test follows the same guidelines as above to compare and verify the routing capacity and the delay introduced by the two message-oriented middlewares in distributing a constantly increasing amount of information among multiple subscribers, when traversing infrastructural and network topologies with different complexity.

The outcome of the test shows (Figure 6) that Zenoh introduces a delay in the delivery of information lower than 3 milliseconds, even when traversing multiple intermediaries. Moreover, the pub/sub protocol implementation realized by these two solutions seems to be less influenced by the complexity of traversed topologies, with a delay variation lower than 1 millisecond.

In conclusion, the tests of this section justify the choice of the authors of introducing Zenoh as a middleware, to retrieve and collect information in arbitrary complex scenarios not only because of its good and more stable performance, even at elevate request rates, but also for its flexibility in supporting multiple interactions paradigms, such as request-reply and pub/sub.

### 6.2. Data Integration

The whole information gathered from actions of Social Sensing, sensors, Providers, and user contributions must pass through a process of adaptation and elaboration so to be accessible and significant for tourists and partners of the pilgrim path. Through the processing phase it is possible to elaborate data coming from actions of social sensing and the different partners to create a customized experience and adapt to variating conditions such as weather, activation of new promotions, or unexpected events.

In this first test of this test-bed section, we aim at showing the effectiveness of the platform components chosen to elaborate and integrate data and services coming from producers. We created a simple function receiving data from either an HTTP endpoint or from a Zenoh Resource and applying a filtering operation and a conversion in JSON format. Transformed data are forwarded through a Kafka topic to Spark where a Spark Streaming will execute some computationally intensive operations on the data received such as square root of the number of occurrences of a character in the data received. The data so processed are then stored into Elasticsearch by the Spark-Elasticsearch connector. To measure the impact of each part on the latency before the data can be stored in the Elasticsearch, each service adds a timestamp in the payload at the receival of any message. We then sent a constant rate of 1000 requests per second and logged the timestamps so generated in Elasticsearch to be visualized through Kibana.

As mentioned before the fast elaboration of gathered data in order to obtain a small end-to-end latency presenting updates in a near real-time fashion to customers is one of the main requirements that emerged from the “Francigena way” use case.

Results shown in Figure 7 show that the platform proposed by authors is able to process and memorize a huge quantity of data and re-exposing them and to provide an extremely limited to provide a very limited total latency of less than 1 s to the customer. The decomposition of the end-to-end latency shows that the sawtooth behavior is due to Spark when processing data with its batch streaming approach.

We can also observe that while the processing delay carried on by the FaaS platform presents less jitter than the processing task by Spark, the many Spark optimizations carried, such as the pre-allocation of computational resources, lead to a processing time one order of magnitude faster. The two major components responsible for the resulting end-to-end latency are the message wait time and FaaS processing time. Messages stored in Kafka queues wait until the Spark adaptor is not ready to receive them in order to be computed and this latency introduced is mainly due to the batch streaming behavior of Spark. The latency introduced by the processing in the FaaS platforms is a known problem of these platforms caused by many factors derived from the dynamic creation of function at each request and is a highly active topic of research [36].

Those results also suggest that while the aim of authors of achieving a less than 1 s delivery time is achieved, when in need of faster end-to-end processing, the tuning of Spark Streaming is the applicable point of intervention. There are, in fact, several options to tune the performance of a Streaming process in Spark, such as increasing the level of parallelism in data receiving and serialization and by setting the right batch interval. Finding the right batch interval (BI), for a Spark Streaming application running on a cluster is an essential condition for its stability and requires that the system is able to process data as soon as it is being received [37]. We have sent a constantly increasing number of requests, starting from 0 to 10,000, during a time-lapse of 5 min and increased the batch interval to test how the infrastructure behaves, at the load increasing.

The variation of BI shows that pipelines configured with a higher batch interval present a higher jitter and latency, while they are more unaffected by variations of loads (Figure 8). In fact, the pipeline configured with a 2 s BI does not show a degraded behavior until the 150 second approximately (with 5000 message/s), while the one configured with a BI of 250 milliseconds shows already a degradation after 75 s (with 2500 message/s) from the test start. On the contrary, if the requirement is a low end-to-end latency, the best configuration is with a BI of 250 milliseconds that can process messages with several orders of magnitude less than other BI configurations.

Zooming in on the test with the BI of 250 milliseconds, we can see that the latency introduced by each part compared with the end-to-end latency is influenced by the processing of the Spark Streaming (Figure 9). These results confirm the necessity of introducing both the FaaS platform and Kafka that guarantee the best flexibility of the infrastructure. In fact, the FaaS platform with its fine-grained scalability has not only exhibited the best adaptation to different connection protocols and message formats, but also the flexibility of performing a preprocessing, before entering the “hard” processing of Spark. Apache Kafka, on its side, is capable of storing information until it is effectively requested by Spark, so as to enable the recovery of transient situations, where the load is excessive to be computed by resources assigned to Spark without losing data.

So, we can claim that a platform, like Apache Spark, can provide a fast, complete, and efficient solution to parallel processing massive amounts of data coming from social sensing, social networks, sensors, and tourism partners.

Altogether, even a tailored tuning of the Spark platform cannot satisfy all different requirements of ST while achieving an optimal usage of computational resources available [38,39]. We can state that Spark is not a solution to manage the heterogeneous fluctuating information of ST scenarios, so the only way to address ST challenging tasks requires to *ad-hoc setup* the Spark infrastructure dimensioning to the worst-case scenario with an obvious waste of resources [40].

On the contrary, the FaaS platform, even with introducing a greater latency in computation, has shown much greater flexibility in response to load variance (Figure 9). Future developments of these FaaS platforms introducing missing computation constructs (such as Map-Reduce) and enabling better performances can shed light to the offloading of the Spark computed part. In this way, it will be possible to leverage on the finer granularity and zero-scaling capabilities of FaaS platforms to grant the best dynamic adaptation to the continuous variations typical of the ST scenarios [36].

From the comparison with partners in the territory, the authors can claim that the proposed first prototype of APERTO5.0 is able to address the challenges in terms of data collecting and fast processing, presented by the creation of a unified platform to support ST services on the “Francigena way”.

## 7. Related Work

To the best of our knowledge general-purpose architecture aiming to integrate and augment different tourist data-sources and services, has already been proposed, however neither as a proposal in literature nor as a business product. However, some work in the literature has formerly proposed infrastructures to support the development of smart tourism services.

RADON is an EU-funded project aiming to develop a model-driven DevOps framework capable of managing the entire lifecycle of services developed and deployed on top of FaaS platforms. The scope of this project also includes the porting of ADAMO, an application that combines tourist preferences with a city mobility network and points of interest in order to generate customized routes [41].

In [42], authors describe a model architecture for smart tourism systems (STS) tailored for cultural heritage and territorial data. The proposed architecture prompts a rethinking of the key paradigms: interactive travel, tourist gaze, hospitality, authenticity, and social networking data owing to the exploitation of software as a service platform tailored on the development of ST services.

However, the proposed platforms, while providing valid support for the development of new services, do not address the problem of heterogeneity of source in data and services nor propose an integration pattern to integrate with existing tourism offer.

In [43], the authors proposed a suite of small applications in tourism, by using a recommendation approach and supported in a microservice pattern, via a set of independent deployable services. The authors define the following functions: (1) to suggest routes and point of interest to users with respect to the choice of tourism activities; (2) the mutual supporting tourist and tourism services suppliers with location services; (3) to help communities in the preservation and valorization of cultural heritages; (4) to enable tourist to share their travel experiences to help other travelers in their decision-making process.

The proposed architecture of APERTO5.0 instead does not focus only on single tourism services or vertical tourism experiences but aims to integrate and involve as many data sources as possible to create a richer, more integrated, and unified tourism experience.

## 8. Discussion and Conclusions

In conclusion, the novel development opportunity of smart tourism via a pervasive redesign due to modern IT technologies constitutes a unique occasion toward more sustainable, inclusive, and culturally rich tourism tailored according to customer preferences. In this context, the cooperation with all stakeholders of the territory and the stimulation of local action of social and sensor sensing are necessary to consider and provide an unprecedented big data amount to enable and stimulate even more the creation of smarter and more connected tourism services. However, the deep fragmentation of services and technologies adopted by different actors in tourism that characterize also the whole information provided by customer sensing and IoTs heterogeneity deeply clash with an effective organization of smart tourism.

It is the authors’ opinion that the introduction of APERTO5.0 can provide a significant contribution to Academia and Tourism business development and management. This paper has proposed APERTO5.0 as an architecture aiming to address the problem of heterogeneity by providing a unifying view in which any tourist item (data, service, and agents) can become part of an integration mosaic capable of accommodating any new possible element. Such an unifying environment is the authors’ main design goal and represents a major innovation in academic research for its novel model and the innovative technological solutions employed in its first proposed prototype.

In fact, the introduction of APERTO5.0 as support to the structuration of ST services over the “Francigena way” allowed us to underline the great adaptability in providing a unifying support over the huge amount of information available. In this context, this solution introduces an innovative mechanism to query and gather data coming from complex scenarios in an efficient and scalable way enabling the introduction of actions of social sensing and partner involvement. A FaaS layer handles gathered data, to cope with the variety and availability fluctuations of information and then processed with Apache Spark. Results showed that the proposed platform is able to collect and process information in parallel, also traversing complex infrastructure topologies, with a resulting end-to-end latency lower than 1 s.

Considering the contribution of APERTO5.0 to the tourism management field, owing to the cooperation with realities of the territory and the explored use-case of tourism paths, authors already demonstrated the effectiveness of proposed solutions. However, a better exploration of the potentiality of a pervasive application of concepts and possibilities opened by Aperto5.0 in the many possible tourism facets and in particular in tourism management deserves better exploration by field experts.

As future directions, authors will work both in the use case of pilgrim’s paths and in new technical widening directions.

Along the first line, through collaboration with tourism realities in the territory, we aim to further develop APERTO5.0 to support novel innovative and integrated forms of tourism. The integration of also other forms and use cases of tourism can increase furthermore the value of the interconnection of services and information toward a unified and pervasive support to ST development.

To follow novel technical directions, we plan to exploit better the potential of serverless infrastructure in providing fast fine-grained scaling of resources and in reducing the time-to-market development of new services. To pursue these capabilities, future research work will focus on optimizations and the introduction of new capabilities in the FaaS platform, so as to easily meet the needs of ST use cases. The main idea is to explore a more pervasive integration of this model of cloud computing with decentralized deployments over the so-called cloud continuum [41] and exploit data and service locality to achieve lower latencies and finer-grained customization of the platform. Moreover, we stress that a fast diffusion of the FaaS is highly likely, so many other features will be available very soon for even better performances.

## Figures and Tables

**Figure 1 sensors-22-01619-f001:**
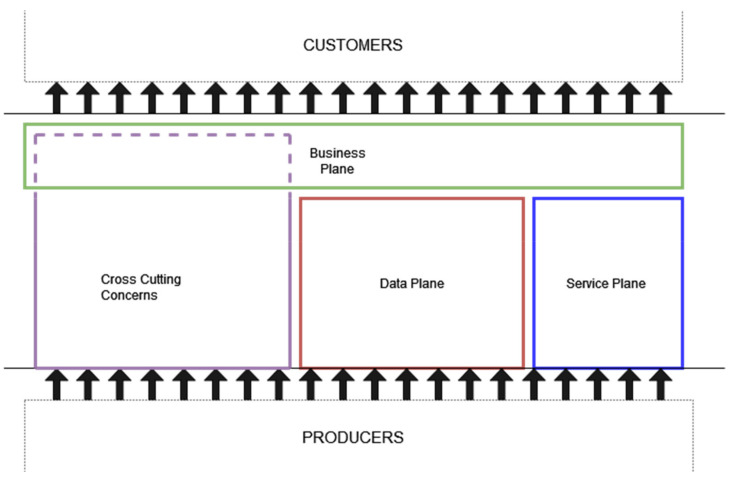
High-level vision of the APERTO5.0 architecture in terms of the layers connecting providers to customers.

**Figure 2 sensors-22-01619-f002:**
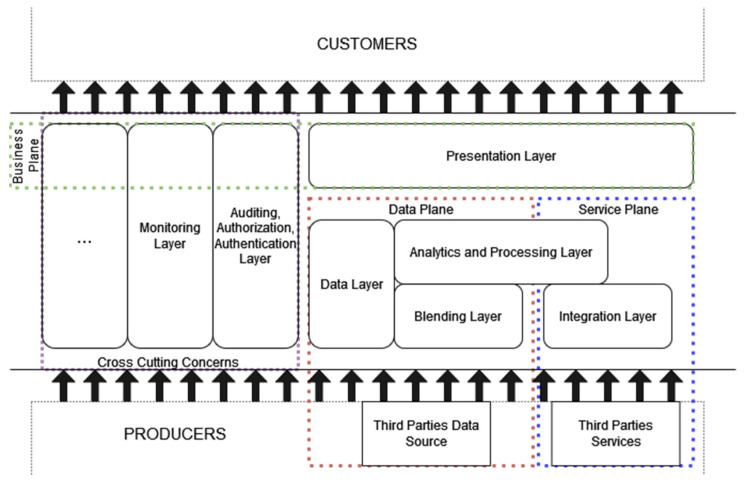
APERTO5.0 layers more detailed view. All components are put together for a more comprehensive effort of integration and synergic coordination.

**Figure 3 sensors-22-01619-f003:**
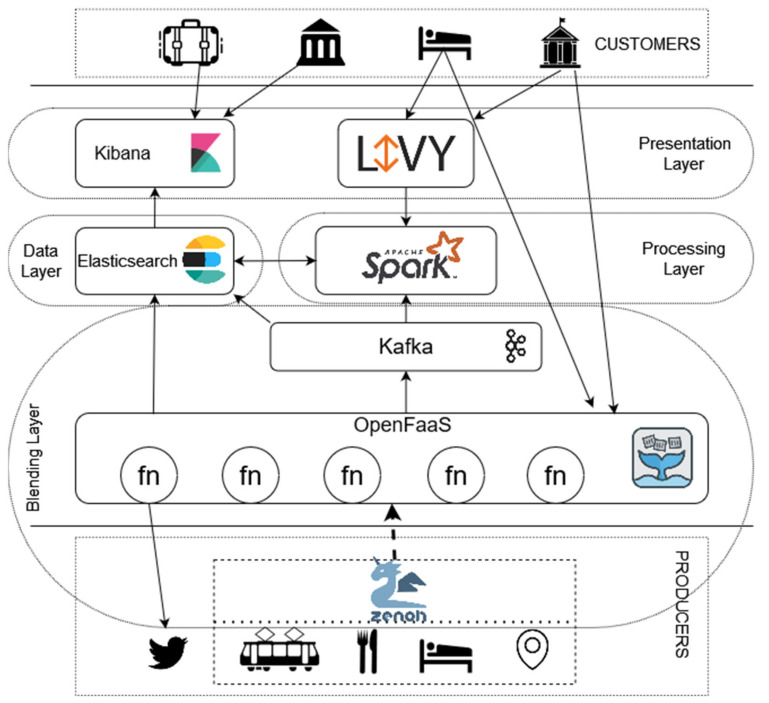
First prototype of APERTO 5.0 realized to satisfy pilgrims path needs.

**Figure 4 sensors-22-01619-f004:**
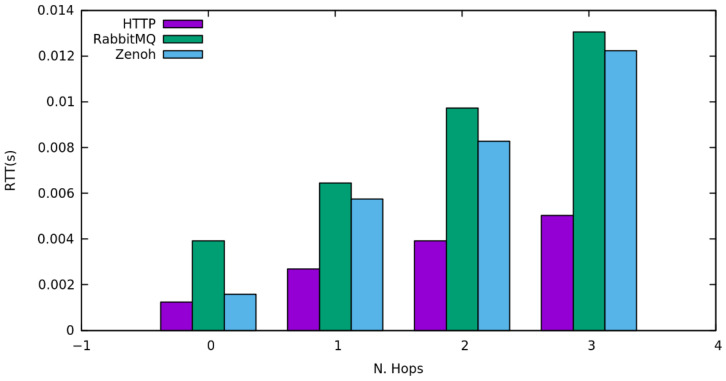
Round trip time of requests when traversing an increasing number of network hops.

**Figure 5 sensors-22-01619-f005:**
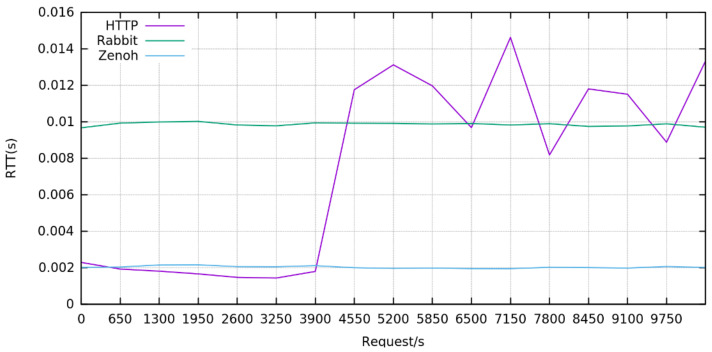
Round trip time of requests when traversing 2 hops with an increasing number of requests per second, starting from 0 up to 10,000.

**Figure 6 sensors-22-01619-f006:**
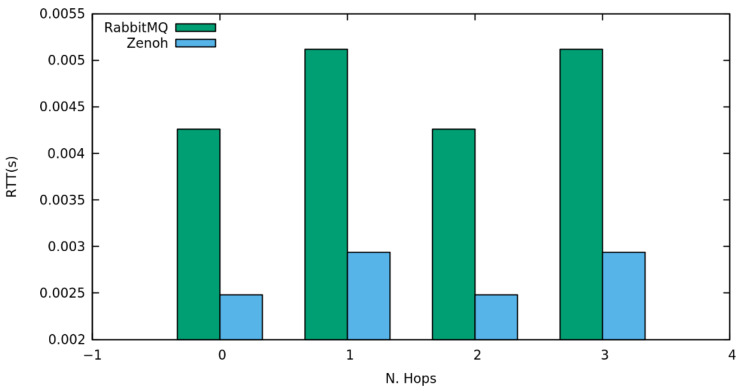
Registered delay in publish/subscribe mechanisms when traversing an increasing number of network hops starting from direct point to point communication to three intermediaries.

**Figure 7 sensors-22-01619-f007:**
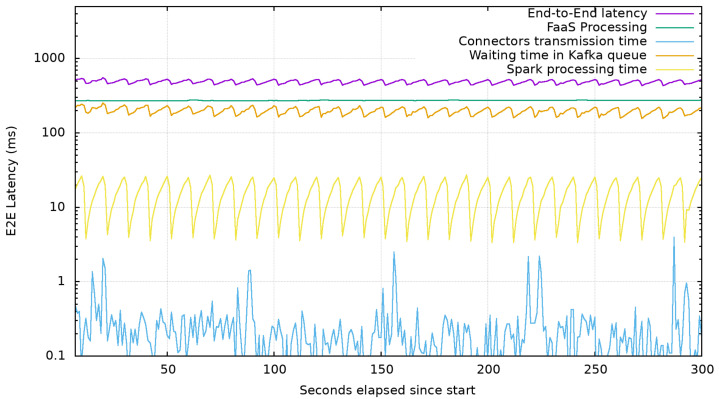
Average latency introduced by each component compared with the end-to-end latency when under a constant load of 1000 message/s (Log Scale).

**Figure 8 sensors-22-01619-f008:**
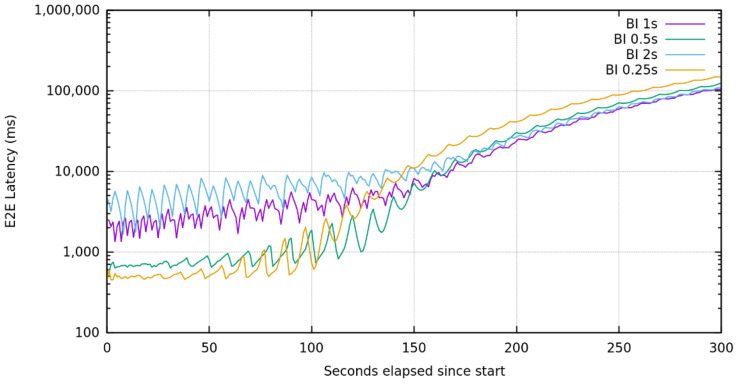
Average end-to-end latency in logarithmic scale when variating the batch interval (BI) in Spark Streaming and submitting an increasing number of messages from 0 to 10,000 in a time-lapse of 5 min.

**Figure 9 sensors-22-01619-f009:**
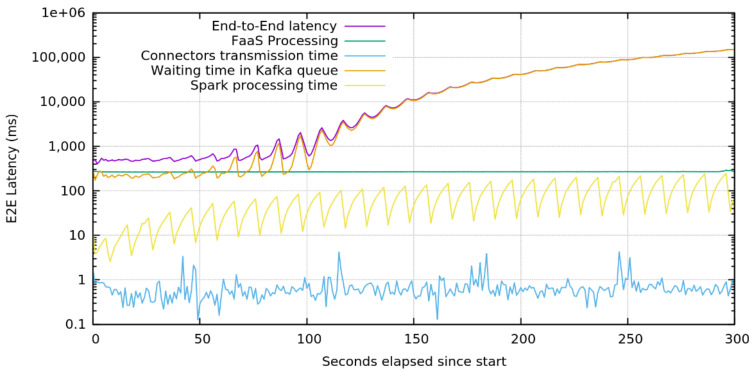
Average latency introduced by each part compared with the tail latency when under an increasing load of messages starting from 0 to 10,000 during a time-lapse of 5 min.

## Data Availability

Data available on request due to restrictions.

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
