# Peer review of "An Architecture for Service Integration to Fully Support Novel Personalized Smart Tourism Offerings"

_sensors, 2022, doi:10.3390/s22041619_

Round 1

Reviewer 1 Report

Thank you for giving me the opportunity of reviewing this manuscript. The proposed  APERT 5.0 is a very interesting and useful way of approaching the tourism business. The paper, in my opinion is very well done. I suggest the authors to prepare a "softer" version for the specialists in tourism, more oriented to the business.

One small recommendation - highlight in the introduction the relevance of the project/subject of the paper to the journal.

Author Response

Thank you for your valuable time spent in the process of review of our work and the positive feedback provided. We are very glad that our work has met your interests.
As you and also other reviewers suggested, we are evaluating the realization of a parallel work more oriented to the Business and we are looking for collaborations in that direction.

We also have added a paragraph in the Introduction section that summarizes and clarifies the contributions and innovative aspects addressed by our work.

Best regards.

P.S. follows the major change in the introduction for convenience, other changes are already present in the latest uploaded version of the manuscript.

Introduction from line 138 to 151

In summary, this paper significantly advances the state-of-the-art literature in the field with the original contributions listed below: i) A novel architecture based on Planes (see Section 3: Business, Cross Cutting Concerns, Data and Service) and Layers (Monitoring, Auditing Authorization Authentication, Presentation, Data, Analytics and Processing, Blending and Integration), to abstract the aspects related to the technologies used and group the different tasks. ii) A practical approach to address the problem of heterogeneity and dissemination of information and services in the context of tourism services. iii) An original implementation of the proposal based on Open-Source projects combining well established platforms with cutting-edge technologies. iv) An application of the proposed architecture to the real-use case of ‘Tourism paths’. v) Some in-the-field experimental results for simple deployment cases to show the feasibility of the proposed approach and the efficiency of the implemented architecture.

Reviewer 2 Report

The paper is well written. I really enjoy reading it. However, some issues should be addressed. Since I come from the tourism management field, I would like to see a more focused and nuanced Introduction. The second paragraph (lines 37-41) should be deleted, we do not need proof anymore that tourism is a good thing for the economy.

Please consult the following essential literature concerning smart tourism destinations, that perfectly fit your introduction: 

  1. Gretzel, U., Sigala, M., Xiang, Z., & Koo, C. (2015). Smart tourism: foundations and developments. Electronic markets, 25(3), 179-188.
  2. Cimbaljević, M., Stankov, U., & Pavluković, V. (2019). Going beyond the traditional destination competitiveness–reflections on a smart destination in the current research. Current Issues in Tourism22(20), 2472-2477.
  3. Radojević, B., Lazić, L., & Cimbaljević, M. (2020). Rescaling smart destinations: The growing importance of smart geospatial services during and after COVID-19 pandemic. Geographica Pannonica24(3), 221-228.

Also, please find more references to ground your research in the terms of technology-assisted experiences, human-centric interaction, personalization and problems that usually arise around it.

Please strengthen your section 7. Related work and 8. Discussion and Conslusions by providing at least one paragraph on the theoretical contribution of your work. 

Other improvements should be done:

  • instead of "Our proposal" for the subtitle, please use "The proposal"
  • please rewrite the paper using the expression "the authors suggest, propose..." instead of "we suggest, propose..."
  • I think the APERTO5.0 is not written in Times New Roman font. 

Good luck in revising your work!

Author Response

Thanks for your valuable time spent in the process of review of our work and the positive feedback provided. We are very glad that our work met your interests. Following valuable indications that you and other reviewers have provided, we have strengthened the introduction and conclusion sections highlighting more the contribution of our work in the field and especially for the sensor venue. 
For this reason, we have added a paragraph in the Introduction section in order to clarify and highlight the contributions and innovative aspects addressed by our work. While we recognize that the second paragraph is partially a redundant repetition of the economical relevance of tourism, we believe that this paragraph can help non-expert of the tourism sector to focus better the context of the application of technologies introduced by APERTO5.0 and the relevance of the tourism sector.
Moreover, the second paragraph helps us to introduce the tourism context in the Italian scenario hosting our reference use-case scenario and industrial partners. We have added the work that you suggested as well as others more business and management-oriented suggested by other authors. We would like to especially thank you for your suggestion as we believe that will significantly increase the value of our work. We have consolidated the Conclusion Section by providing a clear explanation of our contributions to the Academy and tourism Management fields. We have rewritten the paper with a less personal form and corrected some typos such as the font of APERTO5.0.

Best Regards

P.S. follows the major changes in Introduction and conclusion for convenience, other changes are already present in the latest uploaded version of the manuscript.

Introduction from line 138 to line 151

In summary, this paper significantly advances the state-of-the-art literature in the field with the original contributions listed below: i) A novel architecture based on Planes (see Section 3: Business, Cross Cutting Concerns, Data and Service) and Layers (Monitoring, Auditing Authorization Authentication, Presentation, Data, Analytics and Processing, Blending and Integration), to abstract the aspects related to the technologies used and group the different tasks. ii) A practical approach to address the problem of heterogeneity and dissemination of information and services in the context of tourism services. iii) An original implementation of the proposal based on Open-Source projects combining well established platforms with cutting-edge technologies. iv) An application of the proposed architecture to the real-use case of ‘Tourism paths’. v) Some in-the-field experimental results for simple deployment cases to show the feasibility of the proposed approach and the efficiency of the implemented architecture.

Discussion and Conclusions from line 770 to line 793

It's author opinion that the introduction of APERTO5.0 can provide a significant contribution to Academia and Tourism business development and management. This paper has proposed APERTO5.0 an architecture aiming to address the problem of heterogeneity by providing a unifying view in which any tourist item (data, service, and agents) can become part of an integration mosaic capable of accommodating any new possible element. Such a unifying environment is authors main design goal and represents a major innovation in academia research for its novel model and the innovative technological solutions employed in its first proposed prototype.

In fact, the introduction of APERTO5.0 as support to the structuration of ST services over the ‘Francigena way’ allowed us to underline the great adaptability in providing a unifying support over the huge amount of information available. In this context, this solution introduces an innovative mechanism to query and gather data coming from complex scenarios in an efficient and scalable way enabling the introduction of actions of social sensing and partner involvement. A FaaS layer handles gathered Data, to cope with the variety and availability fluctuations of information and then processed with Apache Spark. Results showed that proposed platform is able to collect and process information in parallel, also traversing complex infrastructure topologies, with a resulting end-to-end latency lower than 1 second.

Considering the contribution of APERTO5.0 to the tourism management field, thanks to the cooperation with realities of the territory and the explored use-case of Tourism Paths, authors already demonstrated the effectiveness of proposed solutions. However, a better exploration of potentiality of a pervasive application of concepts and possibilities opened by Aperto5.0 in the many possible tourism facets and in particular in tourism management deserve a better exploration by field experts.

Reviewer 3 Report

Dear authors,

First of all, I am glad to have the opportunity to read the article titled “An architecture for service integration to fully support novel personalized smart tourism offerings”, that I have read with great interest.

Undoubtedly, the subject matter addressed in this work is of considerable interest. However, from my humble point of view, the paper has weaknesses.

In spite of these problems, and considering the interest of the subject, I would like to suggest that the authors revise their paper considering the comments offered below. In my view, the revised version should undergo a new assessment process.

Now, I would like to make some comments and suggestions that should always be understood in a positive way and considering that the different observations constitute different avenues that may allow improving this interesting research and facilitate its publication and impact in the subsequent specialized literature. With this initial caveat in mind, I would like to make the following observations and recommendations to the authors for their reflection and introduction of the changes they consider appropriate:

From my humble point of view, the manuscript should be improved in three aspects:

1) Introduction section:

- The authors should further clarify what the contribution of the paper is, what is new in this paper? Why should it be published? What is the literature gap covered by this paper? what is the associated interest of this contribution? Has anyone previously suggested the need and interest in developing this specific contribution?

2) Discussion and Conclusions section:

- The discussion section is very poor, when it should be the most important part of the article. The authors should include in the discussions section, on the one hand, discussions for the academy, on the other hand, discussions for the management, and another section with the limitations of their study, to end with a conclusion indicating the most important contribution of this research.

3) Final recommendation:

- Additionally, the authors should update the references since there is only one from 2021 and one from 2020. Therefore, I recommend that you include more current references related to "smart tourism" in different markets to enrich your manuscript in this way. And I recommend that you incorporate at least the following references. And I recommend that you incorporate at least the following four current references that have already been cited by other authors. These references could be included in the introduction or in the discussions section:

2020: Smart tourism destinations: a critical reflection. https://doi.org/10.1108/JHTT-01-2019-0011

2020: Sustainable, Smart and Muslim-Friendly Tourist Destinations. https://doi.org/10.3390/su12051778

2021: Travelers' Perception of Smart Tourism Experiences in Smart Tourism Destinations.  https://doi.org/10.1080/21568316.2020.1798689

2021: How smart is e-tourism? A systematic review of smart tourism recommendation system applying data management. https://doi.org/10.1016/j.cosrev.2020.100337

For all these reasons, from my humble point of view, I think this paper need a mayor revision, but I hope that the authors can do it.

Author Response

Thank you for your valuable time spent in the process of review of our work and the positive feedback provided. We are very glad that our work has in some way met your interests.

Following your comments, we have modified the paper as follow:

1) In order to underline more the unique contribution of our work, we have added a final paragraph in the Introduction synthetizing the novelties introduced by APERTO5.0 and underling the specific contribution for the Sensors venue. The comparison with other works available in the literature addressing the theme of Smart Tourism services and information fragmentation and geographical distribution are left in the Related work section.

2) While we have structured the paper with a strong technical orientation, we recognize the value in the distinction of APERTO5.0 contributions over the different fields touched. For this reason, we expanded the conclusion section by differentiating the contribution of our proposal for Academy, with an especial focus on technological innovation introduced, and Business Tourism management. We have also introduced a discussion over the theoretical tourism management aspects that we have not explored even if it deserves deeper research.

3) Following valuable indications provided by you and other reviewers, we have strengthened work references by including:
i) essential literature concerning smart tourism destinations,
ii) newer works addressing the evolution of smart tourism in recent days especially since the covid-19 pandemic outbreak,
iii) and more business and management-oriented literature contributions.
We would like to thank you for your suggested references belonging to more business-oriented venues that we did not consider in our preliminary work.

Best regards

P.S. follows the major changes in Introduction and conclusion for convenience, other changes are already present in the latest uploaded version of the manuscript.

Introduction from line 138 to line 151

In summary, this paper significantly advances the state-of-the-art literature in the field with the original contributions listed below: i) A novel architecture based on Planes (see Section 3: Business, Cross Cutting Concerns, Data and Service) and Layers (Monitoring, Auditing Authorization Authentication, Presentation, Data, Analytics and Processing, Blending and Integration), to abstract the aspects related to the technologies used and group the different tasks. ii) A practical approach to address the problem of heterogeneity and dissemination of information and services in the context of tourism services. iii) An original implementation of the proposal based on Open-Source projects combining well established platforms with cutting-edge technologies. iv) An application of the proposed architecture to the real-use case of ‘Tourism paths’. v) Some in-the-field experimental results for simple deployment cases to show the feasibility of the proposed approach and the efficiency of the implemented architecture.

Discussion and Conclusions from line 770 to line 793

It's author opinion that the introduction of APERTO5.0 can provide a significant contribution to Academia and Tourism business development and management. This paper has proposed APERTO5.0 an architecture aiming to address the problem of heterogeneity by providing a unifying view in which any tourist item (data, service, and agents) can become part of an integration mosaic capable of accommodating any new possible element. Such a unifying environment is authors main design goal and represents a major innovation in academic research for its novel model and the innovative technological solutions employed in its first proposed prototype.

In fact, the introduction of APERTO5.0 as support to the structuration of ST services over the ‘Francigena way’ allowed us to underline the great adaptability in providing a unifying support over the huge amount of information available. In this context, this solution introduces an innovative mechanism to query and gather data coming from complex scenarios in an efficient and scalable way enabling the introduction of actions of social sensing and partner involvement. A FaaS layer handles gathered Data, to cope with the variety and availability fluctuations of information and then processed with Apache Spark. Results showed that proposed platform is able to collect and process information in parallel, also traversing complex infrastructure topologies, with a resulting end-to-end latency lower than 1 second.

Considering the contribution of APERTO5.0 to the tourism management field, thanks to the cooperation with realities of the territory and the explored use-case of Tourism Paths, authors already demonstrated the effectiveness of proposed solutions. However, a better exploration of potentiality of a pervasive application of concepts and possibilities opened by Aperto5.0 in the many possible tourism facets and in particular in tourism management deserve a better exploration by field experts.

Round 2

Reviewer 3 Report

Dear authors,

First of all, I am glad to have the opportunity to read again the article titled “An architecture for service integration to fully support novel personalized smart tourism offerings”, that I have read with great interest.

Undoubtedly, the manuscript has been improved in many ways. However, from my humble point of view, the paper could get better if the authors had included all the references that I recommended in my previous review (and not only one):

2020: Smart tourism destinations: a critical reflection. https://doi.org/10.1108/JHTT-01-2019-0011

2020: Sustainable, Smart and Muslim-Friendly Tourist Destinations. https://doi.org/10.3390/su12051778

2021: Travelers' Perception of Smart Tourism Experiences in Smart Tourism Destinations.  https://doi.org/10.1080/21568316.2020.1798689

For this reasons, from my humble point of view, I think this paper need a minor revision, but I hope that the authors can do it.

Author Response

Dear reviewer

thank you again for your time spent in the process of review of our work.

We have added all the references that you have suggested to us.

Thank you again.

Best regars